# The Potential of Cover Crops for Weed Management: A Sole Tool or Component of an Integrated Weed Management System?

**DOI:** 10.3390/plants12040752

**Published:** 2023-02-08

**Authors:** Margaret Fernando, Anil Shrestha

**Affiliations:** Department of Plant Science, California State University, Fresno, CA 93740, USA

**Keywords:** allelopathy, cover crop termination, roller-crimper, shade, weed suppression, seed bank

## Abstract

Cover crops are an important component of integrated weed management programs in annual and perennial cropping systems because of their weed suppressive abilities. They influence weed populations using different mechanisms of plant interaction which can be facilitative or suppressive. However, the question often arises if cover crops can be solely relied upon for weed management or not. In this review we have tried to provide examples to answer this question. The most common methods of weed suppression by an actively growing cover crop include competition for limited plant growth resources that result in reduced weed biomass, seed production, and hence reductions in the addition of seeds to the soil seedbank. Cover crop mulches suppress weeds by reducing weed seedling emergence through allelopathic effects or physical effects of shading. However, there is a great degree of variability in the success or failure of cover crops in suppressing weeds that are influenced by the cover crop species, time of planting, cover crop densities and biomass, time of cover crop termination, the cash crop following in the rotation, and the season associated with several climatic variables. Several studies demonstrated that planting date was important to achieve maximum cover crop biomass, and a mixture of cover crop species was better than single cover crop species to achieve good weed suppression. Most of the studies that have demonstrated success in weed suppression have only shown partial success and not total success in weed suppression. Therefore, cover crops as a sole tool may not be sufficient to reduce weeds and need to be supplemented with other weed management tools. Nevertheless, cover crops are an important component of the toolbox for integrated weed management.

## 1. Introduction

A cover crop has been defined by the Soil Science Society of America as “a close-growing crop that provides soil protection, seeding protection, and soil improvement between periods of normal crop production, or between trees in orchards and vines in vineyards. When such crops are plowed under and incorporated into the soil, they are referred to as green manure crops” (https://www.soils.org/publications/soils-glossary/# (accessed on 6 February 2023)). Cover crops are generally not harvested but are included in cropping systems because of their documented numerous benefits and, as such, they are considered an important component of sustainable agriculture systems [1]. Among the list of documented benefits of cover crops, weed suppression is often mentioned as one of them [2,3,4]. Extensive reviews of cover crop effects on weed suppression and ecosystem benefits globally have been highlighted in these three papers. However, the question always arises on how reliable or effective a cover crop system is in weed management, and if they can be used as a sole tool or one of the components in conjunction with other methods as an integrated weed management strategy. In this review, conducted by using databases, such as AGRICOLA, AGRIS, BioOne, CAB Direct, PubMed, Web of Science, etc., we synthesized the literature on the success of cover crops in suppressing weeds in agricultural cropping systems and question whether reliance on cover crops alone is sufficient for weed management or whether they are only one of the many tools in integrated weed management systems. The majority of the published studies looking at the weed suppression ability of cover crops seem to have been conducted in North America, Europe, and Australia. 

## 2. Cover Crop Species

Cover crops are generally composed of legumes (*Fabaceae*), grasses (*Poaceae*), brassicas (*Brassicaceae*), and other broadleaf (*Plantago major*) plant families. The optimal plant species for cover crop use depends on the purpose of the cover, the condition of the soil, and the location/climate where it will be grown [5]. Cover crops are chosen based on characteristics, such as ease in establishment, soil coverage, weed and pest suppression abilities, resistance to disease, low competitiveness with the main crop, and ease in termination [6].

The species of cover crop impacts the potential benefits of cover crop adoption. In general, cover crops with high biomass are more beneficial for weed control, soil erosion prevention, and soil organic matter (SOM) buildup. However, species that produce high biomass could also cause competition with the cash crop for resources, such as nutrients, light, and water [6]. In fact, one of the reasons for the low adoption of cover crop systems in semi-arid regions, such as in California, has been cited as water use by cover crops that reduce moisture availability to the cash crop during the growing season [7]. However, recent studies have reported that this may not always be the case [8]. Nevertheless, cover crop benefits may outweigh some of these anomalies in various cropping systems globally.

Cereal rye (*Secale cereale*) is known for its high biomass, ability to compete with weeds, low cost, and winter hardiness, and is one of the most common cover crops grown in maize (*Zea mays*) and soybean (*Glycine max*) cropping systems in the Midwest region of the US [9,10]. Studies from the US and Europe state that legumes, such as hairy vetch (*Vicia villosa*) and balansa clover (*Trifolium michelianum*) are known to be effective in nitrogen (N) fixation and improving the bioavailability of N in soils [11], while grasses and Brassicaceae cover crops are known for nutrient capture [12], while studies in Jordan and Italy have highlighted the allelopathic effect of brassicas and rye on weeds [13,14]. A study in Maryland, US reported that brassica species, such as Siberian kale (*Brassica napus*) and purple top turnips (*Brassica rapa*) reduced the soil compaction by the growth of their taproot system [15].

In some cases, cover crop species are mixed to improve their overall effects [16]. Cover crop mixtures may be beneficial in order to achieve multiple species-specific effects [6]. For example, a study in Australia demonstrated that cover crop mixtures composed of grasses and legumes could increase the SOM through the grass species and increase N fixation and its bioavailability through the legume species [17]. In a study in Atlantic Canada, it was reported that, in general, species mixtures were not more effective in weed suppression compared to monoculture cover; however, when specific highly productive species were mixed, there were benefits in suppression [18].

The location of an agricultural system may impact what species cover crop would be ideal in that system. In cold climates, cover crop species that are winter-sensitive, such as oats (*Avena sativa*), spring triticale (*x Triticosecale*), and clover (*Trifolium* sp.) could potentially reduce water uptake during spring months because they would be frost-killed during the winter. Species that are winter-hardy, such as cereal rye and hairy vetch, have a longer growth period and may result in more water usage in the spring [19]. The water savings effect with winter-sensitive species would not occur in climates not characterized by cold winters because these cover crops would not be frost-killed during the winter. Within a given climate or location, soil conditions, such as soil texture and pH, are important to consider when choosing a cover crop species; hairy vetch is tolerant of low pH soils, while Brassicas grow best in neutral soils [20]. 

## 3. Cover Crops in Perennial and Annual Cropping Systems

Cover crops are more common in the annual cropping systems of the Midwest and Northeast states than in the perennial cropping systems of semi-arid regions, such as those in California [21] and Europe [22]. As mentioned earlier, the depletion of soil moisture by cover crops [7] could be one reason cover crops are less common in the semi-arid climates, however, the differences in management practices between annual and perennial cropping systems may also affect the adoption rates of cover crops in these locations. Because there is a fallow period between the growing season of annual crops, cover crops are often applied to cover bare ground after annual crops have been harvested and before the next set of crops are planted; whereas this is not the case in perennial cropping systems where they are grown in the interrow spaces of the orchards and vineyards. 

In the perennial crops common to California [23], the crops regrow every year and last for many years; when cover crops are used in perennial cropping systems, they are planted between cash crop rows. Perennial systems have a high diversity of management practices and cover crop application depends on the practices used in a specific farming system [24]. There is no set date for planting and termination of cover crops in perennial systems, but often cover crops are timed to reduce in-season competition between the cover crop and the cash crop for growth resources. For example, cover crops were planted in vineyards in California during winter months and terminated in the summer, potentially reducing competition for water during hot summers [25,26]. This practice could reduce the overlap of peak growth stages between the cash crops and the cover crop, and hence the simultaneous demand for resources. In an irrigated vineyard in Spain where *Cynodon dactylon* was a major weed species, a barley (*Hordeum vulgare*) cover crop was successful in suppressing it. The cover crop was planted and terminated in June [22]. 

## 4. Timing of Cover Crop Planting and Termination

As mentioned above, cover crops are usually planted in the fall, early or late winter, or summer depending on the type of cash crop to be planted after cover crop termination in annual cropping systems [7,27,28,29]. In orchards and vineyards, cover crops are usually planted in fall or early winter [8,26,30]. The cover crops are generally terminated before the planting of the cash crop in annual cropping systems or before the orchard crops or grapevines resume active growth after dormancy [30] because the termination date can negatively affect the emergence of cash crops [31] and result in crop yield losses [9,18]. As a general rule, under north American conditions, specifically in the southeast, it was suggested that cover crops should be terminated two to four weeks prior to cash crop planting [32]. Such research-based recommendations generated from studies on the effect of cover crop termination time on trees or grapevines in orchards and vineyards does not seem to exist.

Timing of cover crop planting can have direct implications for the growth rate and amount of biomass accumulation by the cover crop because of a longer growing season, amount of nitrogen fixed in the case of legumes [33], and amount of weed suppression [29,30,34]. For example, it was reported that cover crops produced 40% less biomass, and less nitrogen production by the legumes, when they were planted in mid-October compared to early-September [33]. Haring and Hanson [30] attributed some suppression of weed biomass by cover crops to early planting compared to late planting. However, a longer growing season may also mean more biomass accumulation in both the cover crop and the weeds [28,29]. Studies have also reported that cover crop planting density can also be a factor in weed suppression. For example, Brennan and Smith [34] documented a positive correlation between the amount of weed suppression and cover crop plant density and stated that early-season canopy development by cover crops was important in weed suppression. However, in a study in Australia, it was observed that there was no relationship between cover crop density and weed suppression [35]. Most of the published reports, globally, seem to indicate that suppression of weeds is more in terms of biomass accumulation of the weeds than the density of the weed per se [18,30,36,37,38,39]. 

A study conducted in central Spain concluded that the termination method for cover crops can be critical in optimizing cover crop benefits because it can impact cash crop productivity in annual cropping systems and the ecosystem services from cover crop usage [40]. However, this may not be the case for crop productivity in perennial cropping systems but there are very few studies showing the effect of the cover crop termination method on crop productivity. For example, in an annual crop system in central Italy, a study compared the termination of cover crops at different times with a roller-crimper and glyphosate applications and observed that the sunflower (*Helianthus annuus*) yield was similar between the two systems when the cover crop was rolled late but not when it was rolled at an earlier stage of the cover crop and hence, the authors suggested that early termination of cover crops with a roller-crimper may have to be combined with glyphosate applications [41]. A study compared the chemical, mechanical, and chemical + mechanical termination methods of cover crop termination and their effect on cotton (*Gossypium hirsutum*) emergence and yield, and reported some differences in the crop emergence but no effect on yield [42]. Another study compared two different mechanical methods of cover crop termination on mulch, weed cover and nitrogen but not on crop yield [43]. Kornecki and Kichler [44] compared different cover crop terminations with different roller-crimper types and their effect on the cantaloupe (*Cucumis melo*) yield, but no comparison was made with other cover crop termination methods. However, it can be argued that regrowth of cover crops by an inappropriate termination method could be an issue for crop productivity, especially if the cover crop is still growing actively during the bud break of grapevines in vineyards or the onset of active growth after dormancy in orchards. 

The most common methods for cover crop termination include herbicide application, tillage/incorporation, rolling/crimping, burning, mowing [45], and natural winterkill [46]. Generally, broad-spectrum postemergence herbicides, such as glyphosate, paraquat, glufosinate, 2,4-D, etc., are used for the termination of cover crops [46,47], depending on the type of cover crop. For example, it was reported that glyphosate was effective in terminating cereal rye and wheat (*Tricticum aestivum*) cover crops, but not as effective in terminating legumes. This study further reported that glufosinate controlled legume cover crops, and paraquat plus metribuzin controlled both legumes and cereals effectively, but none of the herbicides when applied alone, or as mixtures, controlled rapeseed (*Brassica napus*) [48]. While herbicide application is a straightforward way of cover crop termination, there are environmental concerns and issues of herbicide resistance that make the sole reliance on this method of termination less appealing [49,50,51]. 

Cover crop residue incorporation with tillage has been shown to be effective in cover crop termination in studies conducted in the US [52] and Denmark [53]; this practice, however, can cause shifts in soil microbial communities and cause damage to the soil structure [54,55,56,57]. Studies from Spain [40] and Italy [58] concluded that cover crop rolling with a roller-crimper was becoming more popular, and this practice enhanced soil health and beneficial biological activity compared to tillage methods. However, rolling methods used alone seem to negatively affect the cash crop yield, as they are less efficient in controlling weed and cover crop populations [52,53]. Studies in Europe have suggested that rolling methods in combination with flaming or herbicide treatments can improve shortcomings of the sole reliance on rolling [40,59]. 

A study in France reported that frost, as a termination method, had benefits on soil characteristics when compared to rolling and herbicide methods of cover crop termination [60]. However, this method would only be effective in climates characterized by cold winters and when using winter-sensitive cover crop species. 

Mowing and tillage are commonly used for cover crop termination in perennial systems [25,61,62]. However, there seems to be less research and publications involving the impacts of different cover crop termination methods in perennial systems likely because cover crop adoption in perennial cropping systems is less common [21,24]. A study reported using flail mowers to terminate cover crops in almond (*Prunus dulcis*) and walnut (*Juglans regia*) orchards in California [30]. Another study reported mowing and allowing the cover crops to senesce as a termination method in vineyards [63]. Perhaps mechanical means of cover crop in orchards and vineyards may be safer than herbicides because the chemicals could drift to the crops and cause phytotoxicity. Usually, as mentioned earlier, cover crops are terminated in spring in orchards and vineyards to avoid competition during the stage when the crops are just resuming active growth after winter dormancy.

## 5. Mechanisms of Weed Suppression by Cover Crops

One of the main goals of cover cropping is to enhance soil health properties but cover crops can aid in weed suppression because of the interactions between the cover crops and weed species. Such interactions that may result in weed suppression could occur during the actively growing phase of the cover crop or after the cover crop dies or is terminated and left as a surface mulch/residue. The various possible interactions are summarized in a conceptual diagram (Figure 1). Plant interactions that aid in weed suppression include direct competition for plant growth resources, allelopathy, facilitation, and indirect interactions [64]. According to the competitive production principle, a species in a shared niche will influence the environment and cause a negative reaction in the other species [65]. Cover crops and weeds may share specific niches in certain cropping systems, causing competition and the suppression of one group by the other. 

Direct competition by manipulation of the seeding rate and method of a rye cover crop was reported to suppress weeds [66]. Biomass and traits, such as plant height, canopy area, and leaf shape also affect the outcome of plant competition [67]. Thus, biomass produced by cover crops can affect light transmittance by creating shaded areas, reduced moisture availability, and reduced soil temperature which in turn can affect the germination of weed seeds [68,69]. Reduced light availability to the weeds in the understory by a taller canopy of subterranean clover (*Trifolium subterranean*) was attributed as a mechanism of weed suppression in a study conducted in the Netherlands [70]. While biomass and leaf area affect competition for light, root length affects nutrient competition [71]. The aboveground plant parts are a direct result of belowground root growth, so plants with rapid root expansion and colonization of root zones are more competitive [72,73]. If cover crops decrease the resource capture of weeds through adjustments to the microclimate, they may out-compete weeds thereby reducing weed pressure in agricultural production systems. Weed suppression by cover crops due to modifications in the soil microclimate has also been reported [74]. Similarly, a study attributed weed suppression in the form of the colonization of weed seeds by bacteria and fungi brought about by soil microbial changes by cover crops [75]. 

While competition is a major mechanism of plant interaction, the physiological properties of cover crops can also influence weed population dynamics; non-competitive interference, such as the chemical interaction of plants, i.e., allelopathy, can cause harm between plant species [76]. It has been reported that allelochemicals produced by certain cover crop species can have a suppressive effect on weeds, and the study documented a linear relationship between allelochemicals produced by a rye cover crop and percent weed inhibition [14]. Several other studies conducted in North America have reported allelopathic weed suppression by a rye cover crop [77,78,79,80,81]. Other cover crop species, such as sunn hemp (*Crotalaria juncea*), cowpea (*Vigna unguiculata*), and velvet bean (*Mucuna deeringiana*) have also been reported to suppress weed germination and growth by allelopathic processes [82]. Similarly, there are several reports of allelopathic weed suppression by sorghum (*Sorghum bicolor*), barley, and wheat [83,84]. Some studies have reported allelopathic effects of the cover crops on the following cash crop [85]. Koehler-Cole et al. [86] published a review on the allelopathic effect of winter cover crops on several row cash crops. Legumes, such as velvet bean (*Mucuna pruriens*), have also been reported to have suppressive effects on weeds in a field experiment in Mexico with corn [87], which perhaps is an example of physical rather than allelopathic suppression. A study in Spain evaluated the allelopathic effects of aqueous extracts from several plant species to explore their potential as a cover crop. Species included *Bromus hordeaceus, B. rubens*, *Festuca arundinacea*, *Hordeum murinum*, *H. vulgare*, *Vulpia ciliata*, *Medicago rugosa*, *M. sativa*, *Trifolium subterraneum*, *T. incarnatum*, *Phacelia tanacetifolia*, *Sinapis alba*, and *Pinus sylvestris* on three weed species *Conyza bonariensis*, *Aster squamatus*, and *Bassia scoparia*. Their results showed differential effects of the extracts in the suppression of the three weed species and concluded that aqueous extracts of some of these species demonstrated that they had potential to be used as cover crops for weed suppression [88].

Indirect interaction between cover crops and weeds includes the cover crop mulch acting as a physical barrier for weed seedling emergence [79] and can also cause shifts in weed populations when cover crops impact the presence of other biocontrol agents, such as omnivorous predators. For example, it was reported that red clover (*Trifolium pratense* L.) cover crops increased seed predation through the increase of predator activity, density, and frequency; the impact of cover crops in this experiment resulted in weed seed removal [89]. 

Depending on the species of cover crop, different plant interactions may occur, affecting different species portions in the weed populations. For instance, crimson clover reduced the eastern black nightshade emergence due to physical suppression, while rye reduced yellow foxtail possibly due to allelochemicals produced by the cover crop [90].

In summary, weed suppression by cover crops seems to be dictated by complex competitive interactions, and the outcomes can often be difficult to predict. A study in France also suggested that competitive outcomes between cover crops and weed species can be due to the complex interaction between resource availability and the traits of the species involved in the competition [91]. Nevertheless, as discussed above, several papers have summarized the major mechanisms of weed suppression by cover crops. 

## 6. Success and Failure of Cover Crops in Suppressing Weeds

The effects of cover crops on weed suppression is highly variable and influenced by many different factors and their interactions. Mainly, in cases where cover crops have been successful in weed suppression, they have been reported to either reduce weed seedling emergence, reduce weed biomass by competing with them, reduce weed seed production, or reduce soil weed seedbanks. The effects could also be a combination of these processes. Although there are more reports of successful weed suppression by cover crops, few studies have reported no effect of cover crops on weeds. 

For example, in a study in an orchard in Turkey, it was reported that living cover crops suppressed weed biomass whereas, mowed and incorporated cover crops reduced weed density [92]. There are several reports of correlation of cover crop mulches with a decrease in weed emergence; however, the species used as mulch influenced the rate of weed emergence [79,93]. A field study that was conducted to assess the effect of residues of rye, crimson clover (*Trifolium incarnatum*), hairy vetch (*Vicia vollosa*), and barley alone and as mixture of all four observed that they reduced the emergence of eastern black nightshade (*Solanum ptycanthum*), while the emergence of yellow foxtail (*Setaria glauca*) was reduced only by rye and barley; hence, suggesting that suppression of emergence not only depended on the cover crop species but also the weed species [90]. Another study compared weed seedling emergence between rye, wheat, and clover residues and observed that while the grain crops suppressed, the clovers stimulated weed seedling emergence [94]. This finding can be explained by a conclusion from a study that cover crop species that contribute to soil nitrogen, such as legumes, may actually stimulate weed seed germination and growth [95]. It has been stated that, during its growth, cover crops reduce both light quantity and quality (red to far red ratio) which in turn will reduce weed seed germination [96]. Therefore, the architecture of the cover crops and the changes it brings about in light quality and quantity may be a factor affecting weed seedling emergence in the case of actively growing cover crops, but there are very few reports of effects of mulch on light quantity and quality, and thereby influence on weed seedling emergence. 

Reductions in weed seedbank sizes are also reported as a weed suppressive effect of cover crops. For example, a study in Italy reported that hairy vetch cover crops reduced weed seedling density, while brown mustard (*Brassica juncea*) showed no effect; the variation in suppressive effects between the cover crop species was not explained by differences in cover crop biomass [97]. A study in Iowa, US reported that winter rye cover crops decreased weed seedbank densities in a maize-soybean farming system. The main weed species affected was common waterhemp (*Amaranthus tuberculatus*) and this study also reported that there was no relationship between cover crop biomass and weed suppression [98]. In contrast, another study in Italy observed a negative relation between these two variables for weed seedbank densities [99]. 

Moreover, reports exist of either actively growing cover crops or mulches having no effect on weed suppression, weed seedling emergence [26,100,101], or decreases in the weed seedbank [102]. The study [100] used rye as a cover crop in a continental climate of Ontario Canada, characterized by hot humid summers and very cold winters and reported that there was no effect on weed density or species composition. Reddy [101] studied the effect of several crops in a humid environment in Stoneville, MS and reported no suppression of barnyard grass (*Echinochloa crus-galli*), prickly sida (*Sida pinosa*), and yellow nutsedge (*Cyperus esculentus*), but some suppression of browntop millet (*Brachiaria ramosa*) densities by Italian ryegrass (*Lolium multiforum*), rye, wheat, hairy vetch, crimson clover (*Trifolium incarnatum*), or subterranean clover cover crops. Baumgartner et al. [26] observed no significant effects of both perennial and annual cover crops on weed populations in a vineyard study that took place in the dry Mediterranean climate with dry summers and mild, wet winters. Other studies also found similar results. For example, in a study in Ontario, Canada, no effect of rye, triticale, and wheat mulches was observed on the emergence patterns of redroot pigweed (*Amaranthus retroflexus*) and common lambsquarters (*Chenopodium album*) [103]. In Oregon, US a study comparing tillage systems with cover crops either lying on the surface or incorporated concluded that tillage type was more important than cover crop mulches in regulating weed seed emergence [104]. Another study in Japan concluded that it was more important to have higher ground coverage at the early stage of the cover crop than using a higher seeding rate of the cover crop for weed suppression [105].

These studies have mostly focused on the suppressive effects or no effect of cover crops on weeds, however, cover crops can also cause increases in weed densities. Cover crops may create shaded areas causing a reduction in weed seed germination, but they can also increase soil moisture, causing conditions favorable to weed seed germination [68]. In one experiment, weed seed densities increased in plots with cover crops, especially in conservational tillage systems, and the composition of weed species was different depending on whether cover crops were present or not [102]. 

In summary, most studies demonstrate evidence that cover crops can influence weed populations; however, many studies have concluded that the effectiveness of cover crops as a form of weed control partially depends on the species chosen as cover crops. The amount of cover crop biomass generally seemed to be positively correlated with reductions in weed biomass and weed seedling emergence. Seasonal differences in weed suppression also seems to be a common theme, for example some studies discussed above mentioned more weed suppression in the spring than in the fall. The overall effectiveness of cover crops seems to also depend on the species of cash crop present, the regional climate, and the condition of the soil. Some examples of these variabilities in weed suppression are presented in Table 1. The combination of positive and negative interactions makes the predicted effect of cover crops on weed populations complicated. Several studies have demonstrated that cover crops alone may not be sufficient to reduce weed densities and need to be supplemented with other weed management tools [30,106].

## 7. Conclusions

Generally, cover crops have been reported to have weed suppressive abilities by influencing weed populations using different mechanisms of plant interaction, which can be facilitative or suppressive. However, the question often arises if cover crops can be solely relied upon for weed suppression. As discussed in this review, there are several variables that affect the success of cover crops in suppressing weeds to conclude that cover crops definitively suppress weeds and can serve as a sole tool for weed management. These factors include cover crop species, time of planting, cover crop densities and biomass, time of cover crop termination, the cash crop following in the rotation, and multiple climatic variables. Further, differences in weed suppression may also vary between annual and perennial cropping systems. Several of the studies also demonstrated that mixes of cover crop species were more successful in suppressing weeds than a single species of cover crop. More research has been conducted in annual than perennial cropping systems. Therefore, future research should explore different cover species which can be used in perennial systems. Most of the species chosen as cover crops in previous studies focused on introduced species, however it could be the case that native species could bring about more benefits due to their acclimation to the environment and their natural ability to compete with invasive plants of the area. Most of the studies that have demonstrated success in weed suppression have only shown partial success and not total success in weed suppression. Therefore, cover crops as a sole tool may not be sufficient to reduce weeds and need to be supplemented with other weed management tools. However, cover crops are an important component of the toolbox for integrated weed management.

## Figures and Tables

**Figure 1 plants-12-00752-f001:**
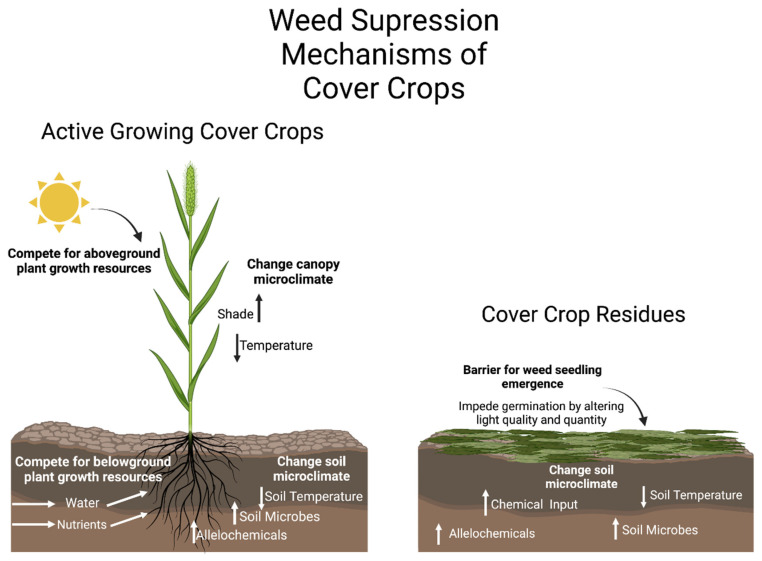
Conceptual diagram of a summary of possible interactions during the actively growing phase of the cover crop (**left**) and after the cover crop is terminated and left as a surface residue on the soil (**right**).

**Table 1 plants-12-00752-t001:** Comparison of the effects of cover crops on weed populations from the literature review.

Cover Crop Species	Background Information	Effect on Weed Population	Reference Nos.
Mulch composed of bark chips, corn (*Zea mays*) stalks, rye (*Secale cereale*), crimson cover (*Trifolium incarnatum*), hairy vetch (*Vicia villosa*), oak (*Quercus* sp.) leaves, and landscape fabric strips	Beltsville, MD Field experiment Silt loam soil	Weed species most affected (from greatest to least) were redroot pigweed (*Amaranthus retroflexus*), common lambsquarters (*Chenopodium album*), giant foxtail (*Setaria faberi*), velvetleaf (*Abutilon theophrasti*)	[79]
Hairy vetch residue	Beltsville, MD Greenhouse study Loamy sand soil	Cover residue reduced velvetleaf, green foxtail (*Setaria viridis*), and common lambsquarters	[68]
Rye mulch	LO, Northern Italy Field study with maize Silt loam soil	Mulch decreased grass and broadleaf weeds by 61% and 96%, respectively	[14]
Rye and sorghum (*Sorghum halepense*)	Petri and Greenhouse experiments Maury silt loam soil	Suppressive effect on barnyard grass (*Echinichloa crus-galli*) due to the allelopathic properties of merced rye sorghum reduced growth of foxtail (*Setaria* sp.) seedlings	[107]
Living cover crops: velvet bean (*Mucuna pruriens*)and Jack bean (*Canavalia* sp.) Mulches: jumbie bean (*Leucaena leucocephala*) and wild tamarind (*Tamarindus indica*)	Xmatkuil, Merida, Mexico Field experiment with corn Clay loam soil	All legumes reduced weed growth with velvet bean	[87]
Red clover (*Trifolium pratense*)	Lafayette, Indiana Field study	Cover crops increased weed seed consumption by the attraction of predators	[89]
Rye, crimson clover, hairy vetch, barley (*Hordeum vulgare*), and a mixture of the four	Field study with plots surrounded by grass and soybeans Silt loam soil	Crimson clover reduced eastern black nightshade (*Solanum ptychanthum*) and merced rye reduced yellow foxtail (*Setaria pumila*)	[90]
A cover crop mix of triticale (*×Triticosecale*), rye, and common vetch (*Vicia sativa*)	Five Points, CA Field with cotton-tomato rotations Panoche clay loam soil	Greater weed seed densities in cover crop plots, especially in conservational tillage systems	[102]
Italian ryegrass, oat rye, wheat, hairy vetch, crimson clover, and subterranean clover	Stoneville, MS Dundee silt loam Soybean field	No effect on densities of barnyard grass, prickly sida, and yellow nutsedge	[101]
Rye cover crops	Delhi, Ontario Loamy sand soil	No effect of cover on weed populations	[98]
Winter rye cover crops	Iowa Maize-soybean field	Decreased seedbank densities, especially common waterhemp (*Amaranthus tuberculatus*)	[96]
Annual cover crops including rose clover, soft brome, zorro fescue, triticale Perennial cover crops including blue wildrye, California brome, meadow barley, red fescue, yarrow	California Vineyard	No effect of cover on weed populations	[26]

## Data Availability

Not applicable.

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
