# Peer review of "The Potential of Cover Crops for Weed Management: A Sole Tool or Component of an Integrated Weed Management System?"

_plants, 2023, doi:10.3390/plants12040752_

Round 1
Reviewer 1 Report
This paper is well written and interesting. It addresses the effect of cover crops on weed suppression in cropping systems. The methodology of the review is not mentioned in this paper. It is noted that there is a tendency to mention works from the USA, however the authors have works from several countries in the bibliography, so you should note that the work does not apply only to the USA (I believe this is the intention of the authors).
The authors present conclusions that are generally consistent with the results, although they do not answer the title question or the last sentence in the introduction, and they should.
The bibliography contains sufficient and appropriate references, but the bibliographic search criteria are unknown.
I leave some suggestions:
Line 38-41: In this review it makes no reference to the methodology used or what criteria were used to do the review. There should be material and methods or at least a brief description of the procedure used in this review
Line 61-68: Convenient to explain the region, we are still talking about the Midwest region of the US
Line 125-126: Again we are assuming we are talking about the northern hemisphere and if we are going to check, specifically South Deerfield, MA in the USA. We must situate the region of the review for better consistency of statements
Author Response
Thank you for the helpful comments. We have addressed your concerns by inserting information on the geographic regions where the studies were conducted. We also state in the introduction that majority of the examples are from North America, Europe, and Australia. We have mentioned how we conducted the review by including the names of the databases we used. Finally, we made some word changes in the abstract and conclusion to address the comment that the title of the paper is not answered in the abstract or conclusion.
Reviewer 2 Report
The topic of the manuscript is relevant. The amount of data is sufficient. However, it needs some revision to provide sufficient information to the reader.
Please, clearly form the research problem in abstract.
You write the keywords
You must write explicitly the objectives of this paper.
The discussion is well developed.
The conclusion should be rewritten and more concertized .
Author Response
Thank you for your valuable comments.
Keywords have been added
Objectives have been mentioned
Abstract and conclusion have been both revised as suggested
Round 2
Reviewer 1 Report
The authors have changed the text according to the suggested revisions, successfully improving the paper. In my opinion, the paper can go forward for publication
Author Response
Although the comments may be valid, we are not able to include these two references because they have nothing to do about weeds and discussing the effects of weeds on variables other than weeds is beyond the scope of our manuscript: Abad et al. about cover crops in vineyards (Abad, J. et al. (2021b). Cover crops in viticulture. A systematic review (2). Implications on vineyard agronomic performance. OenoOne 2, 1-27.) (Abad, J. et al. (2021a). Cover crops in viticulture. A systematic review (1): Implications on soil characteristics and biodiversity in vineyard. OenoOne 1, 295-312.).
Separating annual and perennial cropping systems may be helpful but it is probably not the case in this manuscript as we are only focusing on weed suppression regardless of annual or perennial crops and we are not discussing effect of cover crops in yield of main crop because it is beyond the scope of the paper. Breaking the section into annual and perennial crops does not really add to the manuscript as we discuss these two in separate paragraphs. However, we included the reference Puig et al. (2021) even though cover crops was only one of the many treatments in this study.
We think modifying the structure of the sections does not do much to improve the thought process, discussion, and also does not change anything in the conclusion of the paper. Structure of the paper specially in a review can vary on preferences and both reviewers did not object to this and there was no comment on this either on the first round of review.
We hope the paper is now considered worthy of publication.
